# Hybrid Capture-Based Sequencing Enables Highly Sensitive Zoonotic Virus Detection Within the One Health Framework

**DOI:** 10.3390/pathogens14030264

**Published:** 2025-03-07

**Authors:** Weiya Mao, Jin Wang, Ting Li, Jiani Wu, Jiangrong Wang, Shubo Wen, Jicheng Huang, Yongxia Shi, Kui Zheng, Yali Zhai, Xiaokang Li, Yan Long, Jiahai Lu, Cheng Guo

**Affiliations:** 1School of Public Health, Sun Yat-Sen University, Guangzhou 510080, China; maowy@mail2.sysu.edu.cn (W.M.); jeanwang0626@163.com (J.W.); wujn33@mail2.sysu.edu.cn (J.W.); wjr244209538@163.com (J.W.); zhaiyali1006@gmail.com (Y.Z.); lixk35@mail2.sysu.edu.cn (X.L.); longy73@mail2.sysu.edu.cn (Y.L.); 2School of Laboratory Medicine and Life Science, Wenzhou Medical University, Wenzhou 325000, China; liting@wmu.edu.cn; 3College of Animal Science and Technology, Inner Mongolia Minzu University, Tongliao 028000, China; shubowen@imun.edu.cn; 4Guangzhou Customs District Technology Center, Guangzhou 510623, China; huangjc@iqtcnet.cn (J.H.); shiyx@iqtcnet.cn (Y.S.); zhengk@iqtcnet.cn (K.Z.); 5National Medical Products Administration Key Laboratory for Quality Monitoring and Evaluation of Vaccines and Biological Products, Guangzhou 510080, China; 6One Health Research Center, Hainan Medical University, Haikou 571199, China; 7Research Institute of Sun Yat-Sen University in Shenzhen, Shenzhen 518057, China; 8Key Laboratory of Tropical Diseases Control, Sun Yat-Sen University, Ministry of Education, Guangzhou 510080, China; 9One Health Research Center, Baotou Medical College, Baotou 014040, China; 10One Health Research Center, Wenzhou Medical University, Wenzhou 325000, China

**Keywords:** viral metagenomics, capture probes, one health, viral diagnostics

## Abstract

Hybrid capture-based target enrichment prior to sequencing has been shown to significantly improve the sensitivity of detection for genetic regions of interest. In the context of One Health relevant pathogen detection, we present a hybrid capture-based sequencing method that employs an optimized probe set consisting of 149,990 probes, targeting 663 viruses associated with humans and animals. The detection performance was initially assessed using viral reference materials in a background of human nucleic acids. Compared to standard metagenomic next-generation sequencing (mNGS), our method achieved substantial read enrichment, with increases ranging from 143- to 1126-fold, and enhanced detection sensitivity by lowering the limit of detection (LoD) from 10^3^–10^4^ copies to as few as 10 copies based on whole genomes. This method was further validated using infectious samples from both animals and humans, including bovine rectal swabs and throat swabs from SARS-CoV-2 patients across various concentration gradients. In both sample types, our hybrid capture-based sequencing method exhibited heightened sensitivity, increased viral genome coverage, and more comprehensive viral identification and characterization. Our method bridges a critical divide between diagnostic detection and genomic surveillance. These findings illustrate that our hybrid capture-based sequencing method can effectively enhance sensitivity to as few as 10 viral copies and genome coverage to >99% in medium-to-high viral loads. This dual capability is particularly impactful for emerging pathogens like SARS-CoV-2, where early detection and genomic characterization are equally vital, thereby addressing the limitations of metagenomics in the surveillance of emerging infectious diseases in complex samples.

## 1. Introduction

Pathogen detection technologies play a crucial role in the identification and diagnosis of infectious diseases. Although traditional culture methods are valuable for diagnosing a range of microbial infections, they can be time-consuming and may fail to isolate many pathogens, particularly since a significant proportion of microorganisms are unculturable. Advanced techniques such as polymerase chain reactions (PCRs) and enzyme-linked immunosorbent assays (ELISAs) are effective for detecting known pathogens; however, their ability to identify emerging infectious diseases is limited because they depend on previously characterized pathogen sequences or antigens [1].

In recent years, metagenomic next-generation sequencing (mNGS) has transformed microbiology by enabling the sequencing of DNA from various microbial communities with cultivation-independent and high-throughput methods [2,3,4,5]. This technology has been widely utilized for the rapid and accurate detection of emerging and other clinically important pathogens [6]. However, metagenomic sequencing still faces challenges in numerous scenarios, particularly in detecting samples with low-abundance pathogens, since high ratios of host or background to pathogenic genetic material often hinder the efficient assembly of pathogen genomes, resulting in poor detection sensitivity.

In order to enhance the precision of metagenomic sequencing, enrichment techniques have emerged as a key methodology for the detection of low-abundance pathogens [7]. Currently, targeted enrichment methods include amplicon sequencing and hybrid capture-based target enrichment. Unlike amplicon sequencing, which is limited to predefined regions of interest [8,9], hybrid capture-based target enrichment allows researchers to target a broader range of genomic content, enabling more comprehensive profiling of known and novel variants [6,10]. Thus, it is particularly effective for studying complex and highly diverse pathogen populations [7,11,12].

Zoonotic diseases have increasingly emerged as a critical threat to global public health. A study indicates that 60.3% of emerging infectious diseases between 1940 and 2004 were zoonotic, with 71.8% originating from wildlife. This underscores the significant risk these pathogens pose to human health [13]. Notably, viruses constitute an important component of these pathogens. Moreover, the SARS-CoV-2 pandemic has underlined the necessity of a One Health strategy, which emphasizes interdisciplinary collaboration to effectively address public health challenges.

Given the substantial public health risks associated with zoonotic diseases, the development of a specialized probe library targeting these viruses has become essential. Unlike previous capture-based studies that focus on narrow viral groups or diagnostic targets, our probe library is uniquely designed under the One Health framework, encompassing 663 viruses spanning 41 families, with deliberate prioritization of zoonotic and emerging pathogens. This breadth enables simultaneous surveillance of animal-to-human transmission risks and human-specific pathogens, addressing a critical gap in existing methodologies. In this study, we will compare the virus detection capabilities of uncaptured mNGS and capture sequencing. The establishment of a hybrid capture-based target enrichment deep sequencing system is intended to address a research gap in this field, provide empirical evidence for the One Health concept, and support public health prevention and control strategies.

## 2. Materials and Methods

### 2.1. Selection of Viruses and Probe Design

We selected 663 high-risk viruses capable of infecting humans and animals (including avian species) based on a One Health viral pathogen list [14]. (Detailed information about the viruses is provided in Appendix A). In addition, we included SARS-CoV-2, the novel coronavirus responsible for the outbreak in 2019, using the viral sequence from the strain isolated in Wuhan (accession number NC_045512.2). Relevant viral data were retrieved from the National Center for Biotechnology Information (NCBI) database [15] and categorized according to the classification system established by the International Committee on Taxonomy of Viruses (ICTV, VMR_14-010520_MSL35) [16]. Reference viral sequences were obtained from the GenBank nucleotide sequence database [17].

For viruses with characteristic sequences, probes were designed based on these specific sequences. For viruses lacking characteristic sequences, probes were designed using the full-length reference genome. A sliding window approach was employed to design probes that covered the entire viral genome. The window size was set to 120 nucleotides (nt), with the overlap region determined based on the genome length. Probe design was carried out by Twist Bioscience (South San Francisco, CA, USA), optimizing the melting temperature (Tm) and Guanine–Cytosine (GC) content of each probe. The final probe library consisted of 149,990 probes, covering 41 viral families, with a total size of 18 million base pairs (MB). We used chemically synthesized double-stranded DNA probes from Twist Bioscience, South San Francisco, CA, USA. (The specific genetic sequence information for the probes is detailed in Appendix A).

### 2.2. Viral Reference Samples for Testing

To construct a viral test library, reference samples from four distinct viral species—Severe Acute Respiratory Syndrome Coronavirus 2 (SARS-CoV-2, accession number MN908947.3), Human Influenza A (H3N2, accession number NC_007370), Human Enterovirus 68 (accession number, NC_038308.1), and Measles virus (accession number, NC_001498.1)—were selected. The genomic coverage of the viral nucleic acid reference material exceeded 99.9%, with an initial concentration of 10^6^ copies/μL. A mixture of human nucleic acid reference material (Agilent Technologies, Santa Clara, CA, USA) and viral nucleic acid reference material (Twist Bioscience) was serially diluted to generate viral concentration gradients of 10^5^, 10^4^, 10^3^, 10^2^, and 10 copies/μL. Each viral detection sample contained 25 ng of human nucleic acid reference material. In addition, the negative control consisted solely of human nucleic acid reference material without any virus.

### 2.3. Animal-Origin Samples of Unknown Viruses

Rectal swab samples (n = 14) were collected from bovines from the farm at Inner Mongolia Minzu University in 2022. Swabs were inserted into the perianal region and thoroughly rotated to ensure comprehensive sample collection. Subsequently, each swab was placed in 3–4 mL of a viral transport medium and mixed thoroughly by swirling to release the sample into the medium. The swab was removed and pressed against the side of the test tube to drain the remaining liquid. Used swabs were discarded into disinfectant solutions. Each sample tube was labeled with a unique identifier and stored at −80 °C. All samples were combined into one sample for sequencing.

### 2.4. SARS-CoV-2 Throat Swab Samples

Throat swab samples for SARS-CoV-2 testing were collected by the Guangzhou Customs District in 2022. SARS-CoV-2 nucleic acid (N gene) was detected using real-time fluorescence reverse transcription–polymerase chain reaction (RT-PCR). Samples were classified into three groups based on cycle threshold (Ct) values: low Ct values (Ct ≤ 26, n = 6), medium Ct values (Ct 27–31, n = 3), and high Ct values (Ct ≥ 32, n = 4).

Informed consent was obtained from all participants involved in the study. The study was conducted in accordance with the principles outlined in the Declaration of Helsinki, and the protocol was approved by the relevant ethics committee.

### 2.5. Metagenomic Next-Generation Sequencing and Hybrid Capture-Based Target Enrichment

To begin the experiment, viral nucleic acids were extracted using the Viral Genomic DNA/RNA Extraction Kit (Tiangen Biotech, Beijing, China) according to the HiPure Viral RNA Kit protocol (Magen Biotech, Suzhou, China). The extracted RNA was quantified with a Nanodrop One microvolume spectrophotometer (Thermo Fisher Scientific, Waltham, MA, USA). Whole-transcriptome amplification was then performed on the RNA with the RNA Library Preparation Kit (Reagenex, Lenexa, KS, USA), followed by library construction using the Swift Rapid Library Preparation Kit. Library concentration was quantified with a Qubit 4.0 Fluorometer (Thermo Fisher Scientific) and the Qubit dsDNA High Sensitivity Kit (Thermo Fisher Scientific), while library length distribution was assessed with an Agilent 2100 Bioanalyzer (Agilent, Santa Clara, CA, USA) and an Agilent DNA 1000 Kit (Agilent).

After passing quality control (QC), the libraries were split into two portions. One portion underwent mNGS directly, while the other portion was subjected to hybrid capture-based sequencing, involving liquid-phase hybridization enrichment with capture probes. Hybridization was performed in accordance with the instructions provided by the Twist Rapid hybridization Capture and Enrichment kit (Twist Bioscience). Firstly, libraries were pooled to a total mass not exceeding 1500 ng and combined with probes and blocking reagents, followed by dehydration. Hybridization was carried out at 65 °C. After hybridization, the mixture was bound to pre-equilibrated Streptavidin Binding Beads (Twist Bioscience) and then rigorously washed with pre-warmed Rapid Wash Buffers I and II (Reagenex) to remove unbound molecules. Following capture, PCR amplification was performed using KAPA HiFi HotStart ReadyMix (Roche, Basel, Switzerland), and the amplified product was purified using Agencourt AMPure XP beads (Beckman Coulter, Brea, CA, USA). Finally, the library was quantified with a Qubit 3.0 Fluorometer (Thermo Fisher Scientific) and assessed for quality on an Agilent 2100 Bioanalyzer (Agilent) to ensure integrity.

The reference viral test libraries and those prepared from bovine rectal swab samples were sequenced on the Illumina HiSeq platform at MGI in Guangzhou. Sequencing was also conducted on libraries prepared from SARS-CoV-2 throat swab samples.

### 2.6. Data Analysis and Bioinformatics Pipeline

Reads were aligned to the target viral sequences using BWA-MEM (version 0.7.17) [18] to calculate the number and proportion of reads mapped to the viral genomes.

For the viral standard reference library, the sequences unrelated to the target were removed using the sam2fqMV tool (version 1.19.2). Finally, genome coverage was calculated using samtools (version 1.8) [19] with default view parameters.

For animal samples with unknown viruses, sequencing data from animal samples were first quality-filtered using the Trimmomatic tool [20] and then aligned with the host reference genome using BWA-MEM (version 0.7.17) [18] to remove host-derived sequences. The filtered reads were assembled into contigs using MEGAHIT (version 1.1.2) [21] for viral genome composition analysis. Viral sequences were then identified using BLAST (version 2.9.0+).

For SARS-CoV-2 throat swab samples, sequencing data were processed automatically using the MARS system by Guangzhou Vision Medicals, which included demultiplexing, quality control, removal of low-quality sequences, and adapter trimming. Host-derived sequences were filtered by aligning the cleaned data to a host reference database (IDhost, version 2.0), followed by comparison with a pathogen-specific database (IDseqDB, version 2.0) to annotate taxa at the genus and species levels (IDseqInfo, version 2.0). For SARS-CoV-2 sequences detected in each sample, we analyzed sequencing depth and genome coverage.

## 3. Results

### 3.1. Viral Probe Library

In this study, we selected 663 viruses spanning 41 viral families that are capable of infecting humans and animals (including avian species), corresponding to 1119 reference sequences. The final probe library comprises 149,990 probes with a total size of 18 million base pairs (MB). The taxonomic distribution of the probe library highlights significant representation of viral families such as Flaviviridae, Rhabdoviridae, and Coronaviridae, which are closely associated with emerging infectious diseases (Figure 1). Analysis of genome types revealed that the probe library includes ssRNA(+), dsDNA, and ssRNA(−) viruses, with ssRNA(−) viruses being the most abundant (33.9%), followed by ssRNA(+) viruses (30.6%) (Figure 1). This design ensures compatibility with highly variable RNA viruses.

### 3.2. Test of Hybrid Capture-Based Sequencing in Viral Reference Samples

We performed both conventional metagenomics and probe capture methods on a library of gradient-diluted viral standards to assess the potential of hybrid capture-based sequencing for application in viral genome epidemiology by comparing the ratio of reads to target viral sequences and genome coverage (Figure 2). Viral sequences were detected by both methods in high-concentration viral libraries at 10^5^ and 10^4^, but hybrid capture-based sequencing significantly improved genome coverage and sequencing depth. For example, in the 10^4^ copy number library, the SARS-CoV-2 genome coverage of the uncaptured samples was 31.18%, while the captured samples reached 99.92%. Moreover, at 10^3^ or lower copy numbers, viral detection in uncaptured samples was largely unsuccessful, while captured samples achieved over 60% genome coverage. These results underscore the efficacy of capture enrichment in boosting sensitivity and genome coverage for viral detection.

### 3.3. Application of Capture Sequencing for Identifying Unknown Viruses in Animal Samples

We expanded the evaluation by analyzing calf rectal swab samples to contrast the detection capacity of mNGS with probe capture for unidentified viruses. At the family level, captured samples detected 27 viruses from 11 families, whereas mNGS identified only 11 viruses from 6 families (Figure 3a). For instance, Poxviridae and Circoviridae were exclusively detected in captured samples (Figure 3a). At the species level, conventional metagenomics detected 8 viral species (Figure 3b), while capture sequencing identified a total of 18 species (Figure 3b), including 11 additional viruses (e.g., Feline_stool_associated_circular_virus_KU7) undetected by mNGS. Additionally, we observed that phage sequences were detected using both methods. However, the detection of phages is likely influenced by the high abundance of phages in the sample environment. This could explain their presence in the results, as the abundance of phage sequences in the environment may lead to their unintended capture during the viral enrichment process. Since BeAn_58058_Virus was detected in conventional metagenomics but not in capture sequencing (Figure 3b), we performed further validation by aligning the reads of BeAn_58058_Virus against its reference genome. The results showed that the detection of BeAn_58058_Virus was a false positive, as only 862 base pairs were covered, with a mean coverage of 0.0433 and a coverage percentage of 0.5288%. Given the low coverage and depth, these results indicate that the metagenomic detection of this virus was not a true identification. To quantify enrichment efficiency, we compared the Reads Per Kilobase of transcript per Million mapped reads (RPKM) values of contigs identified by both methods (Figure 3c). The highest RPKM value in the uncaptured group was approximately 10,000 units lower than that of the captured group, with 66.7% of high-confidence contigs (RPKM > 5000) exclusive to enriched data. In suspected viral sequences with greater than 80% gene coverage, only 13 suspected viral sequences were captured in uncaptured samples, whereas 32 sequences were captured in enriched samples through the capture-based method (Figure 3d).

### 3.4. Assessment of Capture Sequencing in SARS-CoV-2 Throat Swab Samples

Throat swab samples from SARS-CoV-2 patients were grouped by Ct values (low: CT value ≤ 26; medium: CT value within 27–31; high: CT value ≥ 32) and analyzed using both conventional metagenomic and probe capture techniques. Although capture sequencing used only one-fourth of the sequencing reads (5 million reads per sample) compared to mNGS (20 million reads per sample), it yielded a substantially higher count of SARS-CoV-2 aligned reads. For medium-Ct-value samples, the number of aligned reads in the captured group was 22 to 55 times greater than in the uncaptured group (Table 1). Genome coverage was also notably superior in captured samples. In high-Ct samples, conventional metagenomics failed to detect SARS-CoV-2, while capture enrichment generated hundreds of mapped reads. These results underscore the enhanced sensitivity of capture sequencing, demonstrating its potential to lower detection thresholds and improve viral detection in low-abundance samples.

## 4. Discussion

Traditional microbial culture methods encounter several limitations, including the premature use of broad-spectrum or prophylactic antimicrobials, the specific growth conditions required for certain species, and their slow growth rates. PCRs and ELISAs are also restricted, which can lead to limited success rates and reduced genome coverage [1]. In contrast, mNGS enables the identification and tracking of diverse pathogens, including low-abundance or novel ones that are difficult to detect by traditional methods [22,23]. This approach not only uncovers pathogen diversity across various environments but also provides a genomic perspective on host–pathogen interactions, improving the prediction and control of zoonotic disease transmission risks. However, mNGS still faces challenges such as interference from host background genomes and contamination from environmental species.

To improve the pathogen detection capabilities of mNGS, various enrichment technologies have been developed on top of it. For instance, hybrid capture-based target enrichment enables the maximization of target reads and aligned reads without the necessity for PCR primer design. This results in a higher degree of mutation accuracy, which makes it an appropriate technique for the rapid diagnosis and monitoring of target pathogens. It is particularly well suited for the detection of low-abundance pathogens in environmental samples, such as wastewater, soil, and air. This provides a reliable means of monitoring potential cross-species transmission. Notable examples of this technology include SeqCap probes (Roche) and VirCapSeq-VERT (Twist) [24].

Established methods like VirCapSeq-VERT have demonstrated remarkable performance in specific use cases, while our approach provides a cost-effective solution for viral surveillance in complex, multi-host environments by designing a probe library that targets both known and potentially emerging zoonotic viruses, which may not be the primary focus of VirCapSeq-VERT. Thus, we provide a method as a low-abundance viral pathogen discovery tool within the One Health framework.

It is worth mentioning that we used chemically synthesized double-stranded DNA probes. Firstly, compared to RNA probes, high-fidelity double-stranded DNA probes, which are quality-controlled through NGS, ensure balanced probe characterization and minimize loss, offering superior performance and advantages in the ease of storage and utilization. Secondly, these probes possess scalability, leveraging probe-shifting technology to ensure optimal performance and precision for viral detection. In addition, we manually validated all viral entries to ensure probe accuracy and eliminated redundant sequences, further refining probe specificity and reducing off-target binding. In this study, we developed a probe set specific to zoonotic diseases and validated its performance using viral reference samples. The results indicated that the detection limit of mNGS in uncaptured samples was approximately 10^3^–10^4^ copies; however, capture sequencing achieved up to 60% genome coverage at comparable concentrations, detecting viral levels as low as 10 copies, reflecting a sensitivity improvement of 2–3 orders of magnitude. Moreover, capture-based enrichment exhibited a distinctive advantage in identifying novel pathogens. For instance, enrichment from animal samples led to the assembly of high-confidence viral contigs, resulting in the discovery of previously uncharacterized viruses [25,26]. Sequence alignment identified a circular DNA virus associated with feline stool (FeSCV) with a genome size of 1301 bp, demonstrating 64.2% coverage and 95.71% similarity to reference genomes. This unclassified circular replication-associated protein-encoding single-stranded (CRESS) DNA virus was first identified in Japan in 2018 [27], with limited data available regarding its genomic characteristics and epidemiology.

For highly mutable RNA viruses such as SARS-CoV-2, capture-based enrichment also exhibited significant advantages [28]. In low-Ct-value samples, the method achieved over 90% genome coverage with minimal sequencing data. In medium-Ct-value samples, detection efficiency improved substantially. Even in high-Ct-value samples, where mNGS struggles to recover viral sequences, capture sequencing detected hundreds of viral reads with genome coverage ranging from 7.60% to 24.47%. However, the efficiency of capture sequencing is critically dependent on the quality and integrity of nucleic acids, particularly for degraded, low-abundance samples.

However, both of these methods have certain limitations. In practical applications, sequences that are abundant in the sample environment, such as phages, may be detected. These detections reflect the challenges of strictly isolating viral targets in environments rich in non-target sequences. Notably, some viruses were detected only in uncaptured samples, highlighting that capture-based methods may miss viruses absent from the probe set. This underscores the need for complementary mNGS in exploratory studies. It is worth noting that while BeAn_58058_Virus is commonly detected in metagenomic sequencing data, our validation process showed that the detection in this study was a false positive. The low coverage and depth of the reads aligned with the reference genome suggest that the results may not accurately reflect the presence of the virus. This finding underscores the importance of validating metagenomic results, especially when dealing with low-abundance viruses, to avoid false positives.

In light of recent updates to the International Committee on Taxonomy of Viruses (ICTV) classification, our probe library design may require periodic updates to reflect new viral taxa and emerging pathogens. Future iterations could focus on optimizing probe design by identifying conserved genomic regions or characteristic sequences, which could reduce the number of probes needed while maintaining or even improving coverage. This approach would enhance the cost-effectiveness and scalability of the probe library, particularly for large-scale surveillance efforts.

## 5. Conclusions

In conclusion, our study presents a hybrid capture-based sequencing method that employs an optimized One Health viral probe set. This method significantly enhanced the detection rates of viral nucleic acids and genome coverage, reduced detection limits, and increased the potential for detecting novel variants within known families. This described method has great potential in zoonotic surveillance within the One Health framework and can be critical for the prevention and control of emerging infectious diseases. Future research should prioritize the optimization of its application across a diverse range of scenarios. Additionally, efforts should be made to explore innovative strategies and methods that can effectively reduce costs associated with its implementation, making it more accessible and feasible for widespread use.

## Figures and Tables

**Figure 1 pathogens-14-00264-f001:**
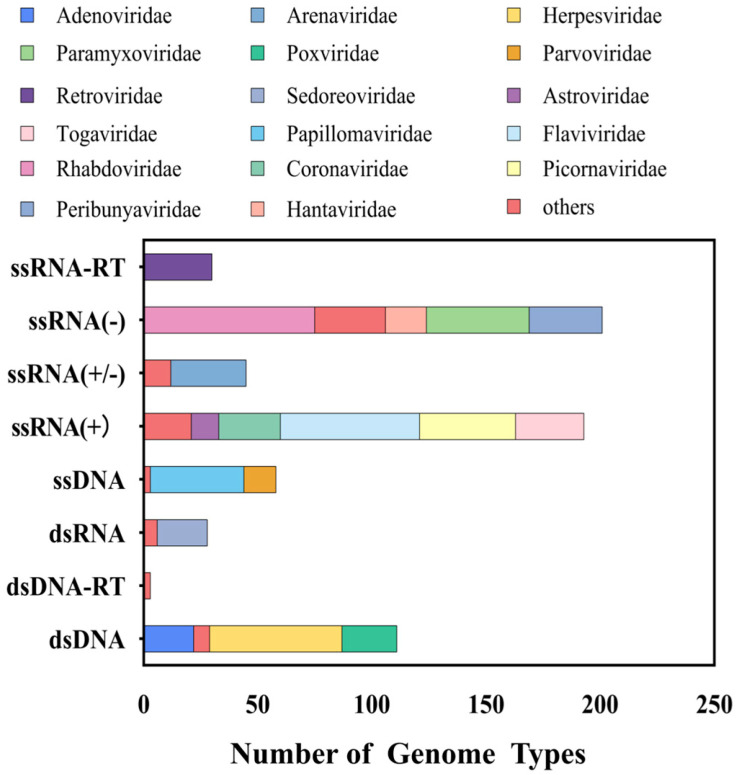
Taxonomic composition of the viral probe library.

**Figure 2 pathogens-14-00264-f002:**
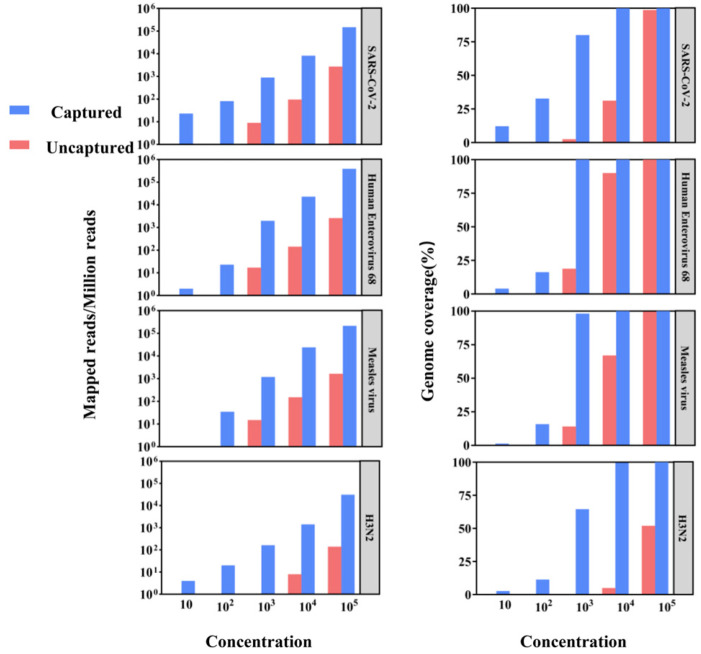
The comparison of aligned reads and genome coverage between conventional metagenomics and viral capture for different viral standards.

**Figure 3 pathogens-14-00264-f003:**
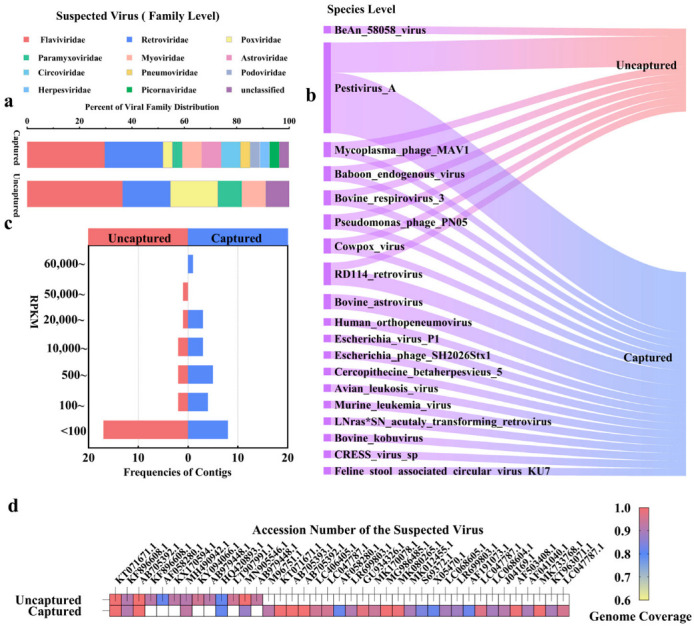
Comparison of mNGS and capture sequencing for detecting unknown viruses in animal samples: (**a**) comparison of viral distribution at the family level; (**b**) comparison of suspected viral species at the species level; (**c**) frequency distribution of the top 25 contigs ranked by RPKM values; (**d**) genomic coverage distribution of suspected viral sequences, with color gradients indicating genomic coverage (yellow: 60–70%; blue: 70–90%; red: 90–100%).

**Table 1 pathogens-14-00264-t001:** Efficiency analysis of capture enrichment for SARS-CoV-2 samples with different viral abundances.

Grouping	Ct Value	Uncaptured Samples		Capture Enrichment Samples	
		Mapped Reads	Genome Coverage (%)	Mapped Reads	Genome Coverage (%)
Low Ct Value	21	10,247	99.11	37,958	99.84
	22	21,082	99.78	57,144	99.84
	24	6718	96.07	27,777	98.18
	25	4114	98.43	41,718	99.19
	25	5359	94.92	36,152	98.87
	25	3647	84.13	23,778	86.43
Medium Ct Value	27	2611	75.43	58,249	88.54
	28	1980	52.02	50,781	91.11
	30	898	12.57	49,459	54.21
High Ct Value	32	0	0	401	24.47
	33	0	0	105	8.11
	34	0	0	251	12.43
	35	0	0	285	7.60

## Data Availability

All data upon which conclusions are drawn are included in the manuscript or in the Appendix A provided.

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
