# Peer review of "Hybrid Capture-Based Sequencing Enables Highly Sensitive Zoonotic Virus Detection Within the One Health Framework"

_pathogens, 2025, doi:10.3390/pathogens14030264_

Round 1
Reviewer 1 Report
Comments and Suggestions for Authors
The manuscript represents an in-solution capture-based method combined with metagenomic sequencing. The method is well established for single species but also for groups of pathogens especially pathogenic viruses. The topic is generally not new and the rationale and the special highlight of this study is not clear to me. The approach should improve the detection of low-abundance viral reads - as other studies already well demonstrated. But is this a diagnostic approach or like the authors to sequence full genomes of viruses?
The existing studies describing this approach for similar compositions of viruses – mentioned in the manuscript – are much better performed and better described with much more details.
The authors of this study claim to present “an optimized One Health viral probe-set” but this optimization is absolutely not demonstrated in the manuscript; it is not clear which parameters are optimized here compared to the already existing capture-based approaches and probe sets described in other papers.
Introduction and methods might be quite okay but the results are only poorly described. There is often no connection between figures and main text. Figures are only poorly explained in legends, graphs are not sufficiently labelled (Fig. 2). The resolution of Figures is not very high.
A critical discussion of the own data is more or less missing, it is rather a repetition of introductory facts.
Special questions/issues:
- Line 28, the mentioned “increase”, is this based on whole genomes or on partial genomes?
- Are the probes based on DNA or on RNA?
- There are not many details given on the composition of the probe set, only families are given but no pathogenic virus species are mentioned. A reference for the “663 viruses” is mentioned but also this does not provide more informative hints about composition. This is completely missing!
- Figure 1 is not mentioned in the text and the part of the text assumingly belongs to this Figure does not sufficiently explain the graph.
- there is also no information on the number of probes per virus and on the completeness of genome which was used for probe design.
- were the probes checked against host genomes?
- in the main text, the authors use captured/uncaptured but the Figures are labelled with enriched/unenriched; I suggest to unify this.
- the description of Fig. 3a (lines 205ff) is misleading and does not coincide with the showed graphs, as I understand. The numbers given in the text do not match with information given in the graphs.
- the description of Fig. 3b and 3d is missing in the results section.
- Fig. 3b, there are given some phage sequences/reads detected in the dataset. However, are probes included in the probe set, which capture this phage sequences; i.e. are these detections a real “capture success” or only sequenced at random when sequencing another part of the library?
- Fig. 3d is poorly explained in the results text, the legend is not explained, what do the numbers/colours mean?
- the missing detections in the “enriched” part (Fig. 3d) need to be discussed. It is important to mention that virus sequences can be lost when applying the capture approach.
- conclusion (see also above), “method significantly enhanced … and genome coverage” (lines 288f), this is not very likely when using only “specific sequences” for probe design (compare lines 93f)!
- conclusion, “increased the potential for discovering novel viruses” (line 290), this is not shown in the study! And it is actually not possible, or only to a certain degree, in my opinion, since probes cannot capture any divergent sequences which would represent novel viruses. If you have investigated this topic, the data must be included in detail in the manuscript.
- some typos should be deleted as well as dashes and the text should be unified for better readability.
Reviewer 2 Report
Comments and Suggestions for Authors
Yet another assay for screening for zoonotic threats under the ‘One Health’ label.
I have no objection to this - but as always the proof will be in the usage - and there are already many different assays for this purposes, so the uptake for this assay may be uncertain.
Reviewer 3 Report
Comments and Suggestions for Authors
In line 84, the authors state, "We selected 663 high-risk viruses capable of infecting humans and animals (including avian species) based on a One Health viral pathogen list." The reference cited does not clearly specify the list of viral pathogens. Please expand on the criteria used to select pathogens for this list.
Line 137: Specify the library preparation kit used.
Line 211: The BeAn 58058 virus is commonly detected in metagenomic sequencing data. However, reads of this virus are often not real detections of this virus. The authors should align reads against a reference genome of BeAn 58058 to verify whether it is a true detection.
Round 2
Reviewer 1 Report
Comments and Suggestions for Authors
The manuscript has improved now and can be published.
Reviewer 3 Report
Comments and Suggestions for Authors
The quality of the manuscript improved significantly after author's review and is suitable for publication.